# Unravelling the Complex Duplication History of Deuterostome Glycerol Transporters

**DOI:** 10.3390/cells9071663

**Published:** 2020-07-10

**Authors:** Ozlem Yilmaz, François Chauvigné, Alba Ferré, Frank Nilsen, Per Gunnar Fjelldal, Joan Cerdà, Roderick Nigel Finn

**Affiliations:** 1Department of Biological Sciences, Bergen High Technology Centre, University of Bergen, 5020 Bergen, Norway; ozlem.yilmaz@hi.no (O.Y.); frank.nilsen@uib.no (F.N.); 2Institute of Marine Research, NO-5817 Bergen, Norway; Pergf@hi.no; 3IRTA-Institute of Biotechnology and Biomedicine (IBB), Universitat Autònoma de Barcelona, 08193 Bellaterra (Cerdanyola del Vallès), Spain; francois.chauvigne@irta.cat (F.C.); alba.ferre@irta.cat (A.F.); joan.cerda@irta.cat (J.C.)

**Keywords:** aquaporin, glycerol, evolution, gene duplication, pseudogene

## Abstract

Transmembrane glycerol transport is an ancient biophysical property that evolved in selected subfamilies of water channel (aquaporin) proteins. Here, we conducted broad level genome (>550) and transcriptome (>300) analyses to unravel the duplication history of the glycerol-transporting channels (*glps*) in Deuterostomia. We found that tandem duplication (TD) was the major mechanism of gene expansion in echinoderms and hemichordates, which, together with whole genome duplications (WGD) in the chordate lineage, continued to shape the genomic repertoires in craniates. Molecular phylogenies indicated that *aqp3*-like and *aqp13*-like channels were the probable stem subfamilies in craniates, with WGD generating *aqp9* and *aqp10* in gnathostomes but *aqp7* arising through TD in Osteichthyes. We uncovered separate examples of gene translocations, gene conversion, and concerted evolution in humans, teleosts, and starfishes, with DNA transposons the likely drivers of gene rearrangements in paleotetraploid salmonids. Currently, gene copy numbers and BLAST are poor predictors of orthologous relationships due to asymmetric *glp* gene evolution in the different lineages. Such asymmetries can impact estimations of divergence times by millions of years. Experimental investigations of the salmonid channels demonstrated that approximately half of the 20 ancestral paralogs are functional, with neofunctionalization occurring at the transcriptional level rather than the protein transport properties. The combined findings resolve the origins and diversification of *glps* over >800 million years old and thus form the novel basis for proposing a pandeuterostome *glp* gene nomenclature.

## 1. Introduction

Prokaryotic and eukaryotic cells utilize glycerol, a three-carbon polyhydric alcohol, as a metabolic intermediate for anaerobic fermentation or aerobic glycolysis, gluconeogenesis, and the biosynthesis of triacylglycerols and phospholipids [1,2]. In addition, due to its colligative properties, organisms as diverse as algae, fungi, insects, and fishes can accumulate glycerol to alleviate osmotic stress or as an antifreeze metabolite [3,4,5,6,7,8]. Though glycerol may passively diffuse across cell membranes [9], its transport is greatly facilitated by a group of integral membrane proteins termed the aquaglyceroporins (Glps) [10,11,12,13,14,15]. Glps were first identified in bacteria as the *Escherichia coli* glycerol facilitator (GlpF), and they have been phylogenetically and functionally shown to belong to a superfamily of transmembrane water-conducting channels, the aquaporins [16,17,18,19].

Subsequent research has revealed a complex evolutionary history of channels capable of facilitating the transmembrane conduction of non-polar glycerol. For example, other members of the aquaporin (AQP) superfamily, including Archaean AqpM, plant GIPs and NIPs, insect Eglps, and vertebrate AQP6, -8, -11, and -14, also evolved this biophysical property [20,21,22,23,24,25,26,27]. In the cases of land plants, which acquired GIPs and NIPs via the horizontal gene transfer of bacterial channels, and hemipteran and holometabolous insects, which evolved more efficient Eglps from mutated AQP4-related channels, the genomic expansions of these new forms of glycerol transporters have been found to correlate to the supplantation of their *glp* genes [13,25,26,28,29]. In nearly all other organisms studied to date, however, the phylogenetically conserved *glps* remain the most ubiquitously selected form of glycerol transporter.

Following the discovery of the bacterial GlpF, multiple Glps were identified in placental mammals (*AQP3*, -*7*, -*9*, and *-10*), which were named according to the chronology of aquaporin gene discovery [30,31,32,33]. In line with this nomenclature, a fifth mammalian Glp, termed *AQP13*, which is phylogenetically related to a frog Glp [34], was identified in the genome of the platypus (*Ornithorhynchus anatinus*), an extant member of the egg-laying prototherian mammals [35]. Conversely, in non-mammalian model organisms, genome-wide studies revealed seven *glp* genes in zebrafish (*Danio rerio*) (*aqp3a*, -*3b*, -*7*, -*9a*, -*9b*, -*10a*, and *-10b*) [36] and thirteen related orthologs in the paleotetraploid Atlantic salmon (*Salmo salar*) [37]. Such gene copy numbers represent ~^1^/_3_ of their genomic aquaporin complements, highlighting their importance for cellular homeostasis. In accordance with the zebrafish information network (ZFIN) nomenclature recommendations, the teleost *glps* were named with respect to their phylogenetic relationships to the mammalian orthologs, although a unified nomenclature for piscine aquaporins has yet to be established.

As for the high aquaporin gene copy numbers in land plants, with up to 120 paralogs in species such as the rapeseed (*Brassica napus*) [38,39], the multiplicity of the teleost and mammalian repertoires has been associated with serial rounds of whole genome duplication (WGD) [40]. For example, in chordates, two rounds (R1 and R2) are considered to have occurred >500 million years ago (Ma), while a third round (R3) occurred ~300 Ma prior to the diversification of Teleostei [41,42]. Subsequently, within the teleost lineage, a fourth round (R4) of autotetraploidization occurred ~80–100 Ma in the common ancestor of Salmoniformes, and allotetraploidization ~12.4 Ma in specific cyprinid lineages [41,42,43,44,45,46,47]. It is nevertheless thought that genome reduction has been the dominant mode of evolution, with duplicated paralogs typically silenced within a few million years [48,49]. As a result, many genes are likely to have been lost, which can obscure interpretations of the origin and modes of diversification of a gene superfamily. Asymmetric gene loss can also have undesired consequences for estimations of divergence times and interpretations of the adaptive functions of the encoded proteins, since it is usually assumed that single gene families represent the same ortholog in each species included in the data set [50,51]. This might be particularly problematic for distantly related piscine lineages that experienced both common and independent WGD and/or gene duplication events, along with the subsequent asymmetric loss of paralogs long after the duplication events. Conversely, the asymmetric acquisition of new genes or the absence of gene differentiation due to concerted evolution may further compound interpretations of historical duplication events.

Amongst more basal lineages of deuterostomes, preliminary studies have indicated that echinoderms such as the purple sea urchin (*Strongylocentrotus purpuratus*) can harbor similar aquaporin and *glp* gene copy numbers to birds, reptiles, and mammals; however, its genome did not undergo WGD [35,52]. How such copy numbers arose and whether they typify the *glp* repertoires of other non-chordate deuterostomes remains to be established.

With the increased availability of public data from large-scale genome sequencing initiatives, it is becoming possible to address such questions in broad groups of distantly related organisms. In the present study, we leveraged >550 genomes and >300 transcriptomes to re-evaluate the evolutionary history of the Glp grade of aquaporins in Deuterostomia using Bayesian and maximum likelihood inference coupled with syntenic analyses. We found that tandem (intrachromosomal) duplication played an unexpectedly important role for *glp* gene evolution and retention in basal lineages, including starfishes, sea urchins, and sea cucumbers (Echinodermata), acorn worms (Hemichordata), lampreys (Hyperoartia), chimaeras and elasmobranchs (Chondrichthyes), as well as spiny ray-finned fishes (Actinopterygii). We assembled and phylogenetically analyzed >100 pseudogenes to validate the proposed evolutionary history, and we experimentally tested the functionality of the channels in a paleotetraploid teleost, the Atlantic salmon, which experienced the most complicated history of gene evolution. Based upon the collated findings, we propose a new pandeuterostome gene nomenclature for the Glp grade of glycerol transporters.

## 2. Material and Methods

### 2.1. Coding Sequence Assembly, Phylogenetic and Syntenic Analyses

Coding sequence (CDS) assembly was conducted as described previously [25,27,35]). Using either full-length proteins or exon-deduced peptides as tblastn queries, open-source whole genome shotgun (WGS), transcriptome shotgun (TSA) and nucleotide databases (NCBI (blast.ncbi.nlm.nih.gov), GenomeArk (vertebrategenomesproject.org), and Ensembl (ensembl.org)) were searched for Glp-related sequences. Full-length proteins were aligned to generate initial multiple sequence alignments using the L-INS-I algorithm of MAFFT v7.453 [53], and the exon/pseudogene-deduced peptides were inserted manually using MacVector (MacVector, Inc., Cambridge, UK). Corresponding nucleotide sequences were then retrieved from the respective DNA contigs or linkage groups (LG) and trimmed to match each peptide fragment before being concatenated to construct the CDS. The data sets selected for phylogenetic analyses were then realigned with the L-INS-I algorithm of MAFFT and converted to codon alignments using Pal2Nal [54]. To detect errors generated by the automated algorithm, test trees were generated via Bayesian inference (Mr Bayes v3.2.2; [55]), and the alignments’ lineage-sorted according to the resulting topology. Identified inconsistencies were then manually corrected using MacVector. Phylogenetic analyses, including Bayesian, maximum likelihood (RAxML 7.2.8 and PAUP v4b10-x86-macosx), and FastTree v2.1.11 [56,57], were performed on the full-length alignments or following removal of the N-terminal exon and gapped regions containing less than three sequences. The Bayesian model parameters were nucmodel = 4by4, nst = 2, rates = gamma for codon alignments, and aamodel = mixed for amino acid alignments. Depending upon the data set, between 1 and 50 million Markov chain Monte Carlo (MCMC) generations were run with three heated and one cold chain, with the resulting posterior distributions examined for convergence and an effective sample size >1000 using Tracer version 1.7.1 [58] and majority rule consensus trees summarized with a burnin of 25%. FastTree and RAxML analyses were conducted using Geneious v9.1.8 (www.geneious.com) with FastTree parameters set to use a generalized time-reversible (GTR) model with rates categories of sites = 40 and an optimized Gamma20 likelihood. The RAxML of codon alignments were conducted with a GTR gamma model using the rapid bootstrapping and search for best-scoring maximum likelihood, (ML) tree algorithm. For amino acid alignments, RAxML analyses were conducted with the CAT BLOSUM62 model. The number of RAxML bootstrap replicates varied between 1000 and 3000, depending on the alignment. ML parameters using PAUP were: heuristic search optimized for parsimony, likelihood GTR model, and NST = 6, with estimated parameters and site rates partitioned by codon. For alignments and accession numbers, see Appendix A. All trees generated were processed with Archaeopteryx [59] and rendered with Geneious. Synteny analyses were partly conducted using the Genomicus v100.01 interface [60], partly using the Ensembl database, and partly via tblastn searches of WGS. Bayesian divergence date estimations were conducted using BEAUTi and BEAST v1.10.4 with relaxed and strict clock protocols. Each run consisted of 100 million MCMC generations of partitioned codon alignments using the GTR substitution model with estimated base frequencies, gamma invariant sites, 4 gamma categories, and the Yule speciation process. Ten normal calibration priors based on the work of Hughes et al. [61] were used in the analyses (see Appendix A).

### 2.2. Experimental Animals and Tissue Samples

Tissue samples were obtained from ~1.5 kg adult males and females of Atlantic salmon held at Matre Research Station of the Institute for Marine Research in Matredal, Norway. Fish were sedated with 10 mg/L of metomidate (Syndel, Victoria, BC, Canada), weighted, and immediately euthanized by decapitation. Tissue samples were taken, immediately frozen in liquid nitrogen, and stored at −80 °C for subsequent total RNA extraction. Procedures relating to the care and use of animals and sample collection were carried out in accordance with the regulations approved by the governmental Norwegian Animal Research Authority (http://www.fdu.no/fdu/).

### 2.3. Cloning of Salmon Aquaglyceroporins

Total RNA was extracted from tissues obtained from four males and four females salmon (21 tissues each; brain, pituitary gland, pineal gland, gills, pseudobranch, skin, muscle, lens, retina, esophagus, stomach, pyloric caeca, mid-intestine, rectum, heart, liver, spleen, head kidney, posterior kidney, ureter, and ovary/testis) using a TRI Reagent^®^ (Sigma-Aldrich, Darmstadt, Germany) following the manufacturer’s instructions. The concentration of the RNA samples was measured on a Nanodrop 1000 Spectrophotometer (ThermoFisher Scientific, Waltham, MA, USA), and RNA integrity was tested using an Agilent RNA 6000 Nano Kit (Agilent Technologies, Santa Clara, CA, USA) on an Agilent 2100 Bioanalyzer and treated with amplification grade deoxyribonuclease I (Invitrogen, Carlsbad, CA, USA) before being used for reverse transcription into cDNA using SuperScript III reverse transcriptase (Invitrogen) according to the manufacturer’s instructions. Approximately 1 µg of total RNA from each tissue sample was used in each reverse transcription reaction.

Salmon aquaglyceroporin transcripts (*aqp3a1a*, *aqp3a2a*, *aqp3b1*, *aqp7_1*, *aqp9a1_v1*, *aqp9a1_v2*, *aqp9b1*, *aqp9b2*, *aqp10aa1*, and *aqp10bb1*) were amplified by conventional PCR from a cDNA library created by mixing cDNAs originating from the above-mentioned 21 tissues of a single male and an ovary. Sense and antisense gene specific primers (GSP) that allowed for the amplification of the whole open reading frames (ORF) for each gene transcript were designed on the 5′ and 3′ UTR from genomic sequences. The list of primers employed in this study is provided in Appendix A. The Easy-A high fidelity PCR cloning system (Agilent) was employed to produce the *glp* transcripts following the manufacturer´s instructions. Obtained PCR products were electrophoresed on 1% agarose gels using GelRed^®^ Nucleic Acid Gel Stain (Biotum) and a TAE-buffer. Target DNA was purified from bands using Monarch^®^ DNA Gel Extraction Kit (New England Biolabs), cloned into the pCR™4-TOPO™ TA vector (Invitrogen), and sequenced using the BigDye™ Terminator v3.1 Cycle Sequencing Kit (ThermoFisher Scientific). Obtained sequences were verified by sequence alignments with the genomic sequences.

### 2.4. Water and Glycerol Uptake Assays in Xenopus laevis Oocytes

Aquaporin constructs in the pCR™4 TOPO^®^ vector were digested with EcoRV/SpeI restriction enzymes and subcloned into the pT7Ts expression vector using T4 Ligase (New England Biolabs). The cRNA synthesis and isolation of *Xenopus laevis* stage V–VI oocytes were carried out as previously described [62,63,64]. Oocytes were microinjected with 1–25 ng of *glp* cRNA in a volume of 50 nl, or with 50 nl of water (control), and incubated at 18 °C over night within a modified Barth’s saline (MBS) medium containing 0.33 mM Ca(NO_3_)_2_, 0.4 mM CaCl_2_, 88 mM NaCl,1 mM KCl, 2.4 mM NaHCO_3_, 10 mM HEPES, and 0.82 mM MgSO_4_ at pH 7.5). The following day, the osmotic water (*P*_f_) and glycerol (*P*_gly_) permeabilities were measured from the time course of osmotic oocyte swelling in a standard assay at pH 7.5 and 8.5. To determine the *P*_f_, oocytes were transferred from 200 mOsm MBS to a 20 mOsm MBS medium at room temperature, and the swelling of the oocytes was followed by video microscopy using serial images at 2 s intervals during the first 20 s period. The *P*_f_ values were calculated by taking into account the time-course changes in relative oocyte volume (*d*(*V*/*V*_o_)/*dt*), the oocyte surface area (*S*), and the molar volume of water (*V*_w_ = 18 cm^3^ mL^−1^) using the formula *V*_o_(*d*(*V*/*V*_o_)/*dt*)/(*S*´*V*_w_(Osm_in_–Osm_out_)). To measure the *P*_gly_, the oocytes were transferred to an isotonic solution containing 160 mmol L^−1^ of glycerol and complemented with 40 mOsm MBS to adjust the solution to 200 mOsm. The *P*_gly_ was calculated from oocyte swelling using the equation (*d*(*V*/*V*_o_)*dt*)/(*S*/*V*_o_) [65]. A slight shrinkage of the control eggs was observed as a consequence of a slight deviation from isotonicity of the incubation medium. Data were plotted and statistically analyzed with an unpaired Student’s *t*-test using the GraphPad Prism v8.4.2 (679) software. A *p*-value < 0.05 was considered statistically significant.

### 2.5. Tissue Expression Profiles of Salmon Aquaglyceroporins

Reverse transcriptase PCR (RT-PCR) was used to determine the tissue expression profiles of each salmon *glp* transcript, and cDNA libraries from 21 tissues from four males and four females were screened for this purpose. cDNA libraries, which were prepared as indicated above, together with the primers listed in Appendix A, were used for the RT-PCR screening. Five-fold diluted cDNA was used as a template for a total of 12.5 µl final PCR reactions in the presence of a 5X Green GoTaq^®^ Reaction Buffer (Promega, Madison, WI, USA), MgCl_2_ (25 mM) dNTPs (1.25 mM each) sense and antisense primers (25 µM each), cDNA, and GoTaq DNA Polymerase (Promega). The PCR cycle conditions were: 2 min of initial denaturation at 95 °C, 35 cycles of 30 sec denaturation at 95 °C, 30 sec annealing at 61 °C, and 1 min 30 sec of extension at 72 °C, as well as an additional final extension for 7 min at 72 °C. These conditions were slightly modified for transcripts that were targeted for detection with lower size amplicons (*9a1_v1* and *9a1_v2*). Ten microliters of PCR products were run on 1% agarose (2% agarose for *9a1_v1* and *9a1_v2*) gel electrophoresis as mentioned above for the visualization of the results on UV-light gel doc imaging system (Biorad, Hercules, CA, USA). The sizes of targeted amplicons were compared to corresponding size range molecular weight markers (GeneRuler DNA Ladder Mix or GeneRuler 50 bp DNA Ladder, Thermo Fisher Scientific). An Atlantic salmon elongation initiation factor 1a (*eif1a*) was employed as a reference gene.

## 3. Results

### 3.1. The Polyphyly of Deuterostome Glps

The first *glp* data set consisted of the aligned codons from 249 taxa assembled from the two major clades of deuterostome organisms, the Chordata (including lancelets (Cephalochordata), sea squirts (Tunicata), jawless (Cyclostomata) and jawed (Gnathostomata) vertebrates), and the Ambulacraria (including starfishes, sea cucumbers, and sea urchins (Echinodermata)), and acorn worms (Hemichordata). Both probabilistic (Bayesian) and likelihood (ML) methods of phylogenetic inference yielded very similar topologies, showing five major subfamilies of *glps* in vertebrates (*AQP3*, -*7*, -*9*, -*10*, and *-13*), a separate cluster of tunicate and cephalochordate *glps*, and polyphyletic clusters of Ambulacrarian *glps* (Figure 1; see Appendix A for the fully annotated tree). High posterior probabilities (pp) and bootstrap support values (bsv) were obtained for the separation between non-vertebrate *glps*, the vertebrate channels, and the majority (80%) of internal nodes comprised by each lineage but not between the vertebrate subfamilies. In an effort to resolve this issue, we conducted separate phylogenetic analyses of aligned codons and proteins from 209 vertebrate taxa (Appendix A). This yielded robust support (pp = 1.0 and bsv ≥ 82%) for the *AQP3*, -*7*, -*9*, and -*10* subfamilies in the jawed vertebrates (Gnathostomata), as well as the *AQP13* subfamily in prototherian mammals and amphibians, but not for the *glps* identified in the jawless cyclostome hagfishes (Hyperotetri) and lampreys (Hyperoartia).

In hagfishes, we identified two forms of *glps* (CDS: 53% identical), each consisting of six exons. One of the *glps* in the inshore hagfish consistently clustered with the tetrapod *AQP13* channels, while the other *glp* identified in the inshore, Pacific, and Atlantic hagfishes, either clustered together with *AQP3* channels with varying support (pp = 0.56–0.86 and bsv = 100%) or as a polytomy between *AQP3* and *AQP13* channels. Conversely, as in our earlier study [35], we identified four *glp* channels in the genomes of the arctic and sea lampreys, each consisting of six exons (previously named *aqp3L1*, *aqp3L2* (CDS: 72% identical), *aqp10L1*, and *aqp10L2* (CDS: 81% identical); CDS: 56% identical between the *aqp3L* and *-10L* paralogs). Within all data sets, the lamprey channels were always placed within two binary subclusters with highly robust support values (pp = 1.0 and bsv = 100%). The *aqp3L* binary subcluster always clustered on the same branch as the gnathostome *AQP3* channels (proteins: pp = 1.0 and bsv = 100%; codons: pp = 0.5–0.75 and bsv = 50–100%), while the previously named *aqp10L* binary cluster either clustered on the *AQP9* branch of vertebrate *glps* (pp = 0.6 and bsv < 50%), on the same branch as the tetrapod *AQP13* channels (pp ≤ 0.86 and bsv < 50%), or as a polytomy between *AQP3* and *-13* channels. Further analyses in a separate data set of 163 taxa consisting of the cyclostome, chondrichthyan, and amphibian *glps* confirmed the above observations but did not provide higher resolution (Appendix A). These latter studies were conducted to rule out potentially undiscovered *glps*, such as the novel *AQP13* channels in amphibians, but none were identified in the three extant orders—the Anura, Caudata, and Gymnophiona. In addition, none of the cyclostome *glps* displayed any conserved synteny to the vertebrate channels, and we therefore assigned orthologs based upon the phylogenetic analyses conducted in the present study. The hagfish *glps*, which are located on separate contigs, were named *aqp3L* and *aqp13L*, with the “L” representing “like.” Conversely, the lamprey *glps*, which are encoded in three genomic loci, were named as previously *aqp3L1* and *-3L2* on chromosome 16 and 40, respectively, while the previously named *aqp10L* channels were renamed *aqp9_13L1* and *-9_13L2* to reflect their phylogenetic interrelationships with the vertebrate *glps*. Interestingly, these latter channels are tandemly arranged with a short 4.6 kb intergenic region on chromosome 1, indicating that the four *glps* in Hyperoartia evolved through different mechanisms of gene duplication, i.e., interchromosomal and tandem duplication.

### 3.2. Common and Uncommon Duplications in Echinoderms and Hemichordates

Our initial analyses of the non-vertebrate deuterostome *glps* (see Figure 1) revealed at least three polyphyletic clusters (*glp1–3*) in the Ambulacraria and two sister clusters of *glps* in cephalochordates and tunicates. To better understand the origins of these clusters, we assembled CDS from 41 echinoderms, two hemichordates, four cephalochordates, and 11 tunicates, and we conducted extensive phylogenetic analyses of the aligned codons. The data for the Echinodermata encompassed all five classes, including the sea lilies (Crinoidea: one genome), brittle stars (Ophiuroidea: two genomes and two transcriptomes), starfishes (Asteroidea: five genomes and eight transcriptomes), sea cucumbers (Holothuroidea: seven genomes and five transcriptomes), and sea urchins (Echinoidea: four genomes and seven transcriptomes). The results revealed an unexpected relationship between the genomic loci (Figure 2A) and the molecular phylogenies (Figure 2B–D; see Appendix A for a fully annotated tree). Within the genomes of asteroids, such as the European starfish (*Asterias rubens*) and echinoids, such as the purple sea urchin (*Strongylocentrotus purpuratus*), all three types of *glp* (*glp1–3*) are linked, while in the genome of the crown-of-thorns starfish (*Acanthaster planci*), only the *glp1* and *glp3* paralogs are linked, with *glp2* located on a separate scaffold. Conversely, we found multiple copies of *glp1*-type genes in the Echinoidea, Holothuroidea, and Asteroidea, but we found none in Ophiuroidea or Crinoidea. The Ophiuroidea instead evolved multiple copies of the *glp2*-type channels, while our current analysis only revealed a *glp3*-type channel in Crinoidea.

A surprising feature revealed by the synteny and molecular phylogenies was that one of the *glp1*-type channels in Asteroidea, which we named *glp1b*, is co-located just downstream of the *glp3* channels yet shares a close phylogenetic relationship with two tandemly-arranged *glp1*-type channels, which we named *glp1a1* and *glp1a2* in accordance with their coding direction. The loci of these latter channels, which in some species (such as the bat star (*Patiria miniata*)) may include additional pseudogenes, are also tandemly arranged in the genomes of Echinoidea and Holothuroidea, but they are not currently found in Ophiuroidea or Crinoidea. Nevertheless, despite their tandem arrangements in three of the echinoderm classes, the phylogenies showed that gene duplication likely occurred independently. Conversely, the existence of linked *glp1*, *-2*, and *-3*-type channels in three classes of echinoderms suggests a common origin through intrachromosomal duplication, with the potential loss of *glp1* and *-2*-type orthologs in Crinoidea, and the *glp1* and *-3* orthologs in Ophiuroidea. In this latter class of brittle stars, the data further showed that, possibly as compensation for the gene losses, three forms (CDS: 56–66% identical) of *glp2*-type channels evolved (*glp2a*, *-2b*, and *-2c*). Due to insufficient genome assembly levels, it is currently not possible to determine the mode of gene duplication.

Multiple tandem duplications nevertheless seemed to have occurred in the hemichordate acorn worms. In both studied species, *Ptychodera flava* and *Saccoglossus kowalevskii*, either three or five (including two pseudogenes) *glp3*-type genes were identified as tandemly arranged in their respective genomes. All molecular phylogenies showed that they form two sister clusters within a polyphyletic grade of echinoderm *glp3*-type channels. Based upon the loci and coding direction of the *S. kowalevskii* paralogs, we named the channels *glp3a*, *-3b*, *-3c*, *-3d*, and *-3e*. Though the *glp3e* pseudogene was excluded from the analyses due to insufficient sequence data, the phylogeny of the remaining *glps* revealed that *glp3a*, *-3b*, and *-3d* tandemly duplicated independently in each species, but the *glp3c* genes had a common origin. Furthermore, although Bayesian inference (Figure 2B) indicated that the echinoderm *glp1*-type genes are probably the most closely related (pp = 0.78) to the cephalochordate and tunicate *glps*, the FastTree, PAUP, and RAxML likelihood methods of phylogenetic inference indicated that *glp3*- or *glp2*-type channels (bsv = 97%) are likely equally related (Figure 2C,D). Consequently, it seems plausible that a single form of *glp* channel existed in the last common ancestor of the Ambulacraria and Chordata, with molecular features that encompass all three of the major echinoderm forms, and that gene expansion occurred independently in each lineage.

### 3.3. Unexpected Duplications in Jawed Vertebrates

We identified unexpected duplicates in each of the vertebrate *glp* subfamilies. For example, amongst *AQP7*-type channels, the global analysis (Figure 1) showed that tandem duplication occurred in the primate lineage. To better understand this event, we separately analyzed 39 *AQP7*-related primate sequences. This revealed multiple copies of *AQP7* (*AQP7L1* and *-7L2*) in Homininae (gorillas, chimpanzees, and humans) but only a single *AQP7* ortholog in the genome of the Sumatran orangutan (*Pongo abelii*), a member of the great apes (Hominidae), and single *AQP7* orthologs in the genomes of older primate lineages (Appendix A). In all of the great apes, canonical *AQP7* is encoded on Chr 9 just downstream of *AQP3* (Appendix A). The phylogenetic data revealed that this canonical *AQP7* gene tandemly duplicated to form *AQP7L1* in Homininae and then again to form *AQP7L2* in chimpanzees and humans. The syntenic data further showed that *AQP7L1* remained linked to *AQP7* in gorillas and chimpanzees but was translocated to the centromere region of Chr 2 in humans, where it now comprises a pseudogene. Conversely, the second tandem duplicate (*AQP7L2*), which is juxtaposed with *AQP7L1* and thus remains linked on Chr 9 in chimpanzees, was amplified and fractionated along Chr 9 in humans to form the *AQP7p1*–*7p5* pseudogenes.

At the other end of the gnathostome evolutionary time scale, we found tandem duplicates of *AQP3* and *AQP10* orthologs in the genomes of nearly all of the examined chondrichthyan class of fishes (Figure 1). For *AQP3* orthologs, the tandemly arranged duplicates (*aqp3C1* and *-3C2*) were identified in all species and thus arose prior to the separation of the two extant subclasses of chimaeras (Holocephali) and elasmobranchs (Elasmobranchii), while for the *AQP10* orthologs, a second tandem duplication appears to have occurred after the elasmobranch lineage diverged from that of the chimaeras. Though *aqp10* duplicates were found in the transcriptome of the ghost shark (*Callorhinchus milii*), only single copy genes lacking introns were identified in the genomes of the two currently available species, the ghost shark and the small-eyed rabbitfish (*Hydrolagus affinis*). We therefore named the tandemly duplicated *AQP10* orthologs in sharks and rays *aqp10C1* and *-10C2*, while the apparently non-redundant transcripts (CDS: 73% identity) in the ghost shark were named *aqp10C2a* and *-10C2b* to reflect their phylogenetic affinity the downstream *aqp10C2* orthologs.

In past studies of teleost *glps*, the *AQP10* orthologs were found to have duplicated and were named *aqp10a* and *-10b* to reflect their presumed R3 origins [35,36,40,66,67,68]. In the present study, however, we found a more complex heritage of these genes. In the most basal lineages of actinopterygian fishes, represented by the extant bichirs (Cladistia) and sturgeons (Chondrostei), we identified two tandemly arranged *AQP10* paralogs. Amongst holostean fishes, we further identified a partial sequence of the upstream paralog in the spotted gar (*Lepisosteus oculatus*) and complete sequences of these orthologs in the bowfin (*Amia calva*) transcriptome. The phylogenetic analyses showed that the teleost orthologs, previously named the *aqp10a* and *-10b*, respectively, clusters (pp: ≥ 0.98 and bsv: 69–100%) with the upstream and downstream orthologs of the cladistian, chondrostean, and holostean fishes (Figure 1, Appendix A). Subsequent analyses not only confirmed these observations but uncovered the true R3 duplicates in clupeid, salmonid, and gadiform fishes (see “Dependent and independent duplicates in teleosts” below) and revealed that an ancient *aqp10* gene tandem duplication occurred in the common ancestor of all extant actinopterygian fishes. We therefore renamed the pre-teleost *AQP10* orthologs *aqp10aa* and *-10ab* according to their genomic loci and coding direction, and the inherited teleost genes *aqp10aa* and *-10ab*, with R3 duplicates named *aqp10ba* and *-10bb*. In addition to the ancient tandem duplication at the root of Actinopterygii, our data revealed independent tandem duplications of the *aqp10bb* orthologs in Pacific and Atlantic eels (Anguillidae) and the *aqp10aa* orthologs in a suborder (Cyprinodontoidei) of the tooth carps (Cyprinidontoformes). Amongst sarcopterygian animals, we found *AQP10* tandem duplicates in select species of amphibians, lizards, and birds, including the two-lined cecaelian (*Rhinatrema bivittatum*); the common wall lizard (*Podarcis muralis*), which also retains a tandemly duplicated *AQP7*; and the golden-collared manakin (*Manacus vitellinus*). Conversely, we confirmed the absence of *AQP10* orthologs in turtles and prototherian mammals, as well as the fractionation of the *AQP10* orthologs in Ruminantia and the mouse (*Mus musculus*) [69,70]. In the latter instance, our searches indicated that gene fractionation appears to have commenced early in the evolution of the Muroidea after the separation of the mole rats (Spalacidae) from the other muroid lineages. Finally we identified two linked *AQP10* pseudogenes (CDS: 100% identical; located at 1.0 and 55.7 Mb of scaffold UNPS02014798) in the metatherian common wombat (*Vombatus ursinus*).

Phylogenetic analyses of the gnathostome *AQP9* channels revealed the least surprises, with single orthologs sorting as expected within the Chondrichthyes (~ 90% pp >0.9; 70% bsv >90%) and Sarcopterygii (100% pp >0.98; 80% bsv >80%) (Figure 1; Appendix A). Lineage sorting was also robust within the Actinopterygii (~ 91% pp = 1.0; 70% bsv >90%), although we found a tandem duplicate in the chondrostean sterlet (*Acipenser ruthenus*) and confirmed the R3 duplicates in teleosts, which showed double-conserved synteny to the holostean spotted gar genes on Chr 3 (see “Dependent and independent duplicates in teleosts” below). We further only identified single *AQP13* orthologs in Amphibia, either assembled from the genomes or expressed in the transcriptomes of Anura (frogs and toads) and Caudata (salamanders), but none in the Gymnophiona (caecilians). In addition, we confirmed the existence of *AQP13* in Prototheria through the assembly of full-length CDS (93% nucleotide identity) from seven exons encoded in the genomes of the platypus and the short-beaked echidna (*Tachyglossus aculeatus*) (Appendix A). We previously considered the platypus *AQP13* ortholog to be a pseudogene based upon the data available at the time [35]; however, although not detected in the platypus transcriptome, the present observations indicated that prototherian *AQP13* genes, which encode 277 amino acid (28.9 kD) proteins, could be functional.

### 3.4. Dependent and Independent Duplicates in Teleosts

To further assess the *glp* gene diversity in actinopterygian fishes, we assembled CDS from >200 genomes and 82 transcriptomes encompassing 53 orders and 118 families. Codon alignments of each subfamily (*aqp3*, *-7*, *-9*, and *-10*) were then analyzed via Bayesian inference and summarized with a majority rule consensus after computing 15–50 million MCMC generations. The lineage sorting of the resultant trees was then compared to the recently proposed species phylogeny [61]. This allowed us to not only delineate the nodes representative of R3 and R4 but also to identify six lineage-dependent duplications that were not associated with either of the fish-specific WGDs (Figure 3A–D; see Appendix A for the fully annotated trees).

The first non-WGD lineage-dependent duplication was detected amongst *aqp3* orthologs, where an interchromosomal duplication (pp = 0.97) occurred in the last common ancestor of the Elopomorpha (eels and tarpons) and Osteoglossomorpha (bony tongues) to generate *aqp3b1* and *-3b2* paralogs in each species (Appendix A). Separate Bayesian inferences (1 million MCMC generations) of the codon alignments of these channels assembled from the genomes of seven-to-eight species of eels and tarpons and seven species of bony tongues in relation to the tandemly duplicated chondrichthyan *aqp3C1* and *-3C2* orthologs (N = 11 species) confirmed (pp = 1.0) that the duplication events in the teleosts and Chondrichthyes occurred independently (Appendix A; see Appendix A for the fully annotated tree). The main tree topology (Figure 3A) also showed that *aqp3a* paralogs are absent in Elopomorphs and Osteoglossomorphs, while their *aqp3b1* and *-3b2* channels form a sister clade to the R3-generated *aqp3a* and *-3b* orthologs of all other clupeocephalan teleosts. Syntenic analyses revealed that genes immediately flanking the *aqp3b1* and *-3b2* paralogs display dual synteny to the *aqp3* locus in the spotted gar, as well as the *aqp3a* and *-3b* loci of clupeocephalan teleosts (Appendix A). The *aqp3b2* locus is nevertheless more conserved to the loci of teleost *aqp3a*, spotted gar *aqp3*, and the Chondrichthyan *aqp3C1* and *-3C2* channels, which show conserved syntenies to the sarcopterygian *aqp3* loci (Appendix A). In the latter instance, the downstream juxtaposition of *AQP7* paralogs observed in primates is conserved in all sarcopterygian species but lost in euteleostean fishes. We did, however, detect linked *aqp7* genes at variable distances from the respective *aqp3* or *-3b* paralogs in basal actinopterygian lineages, including the reedfish (*Erpetoichthys calabaricus*: Cladistia), the sterlet (Chondrostei) and clupeid and ostariophysan teleosts (Otomorpha) (Appendix A).

The second non-WGD lineage-dependent duplication was detected in the cod-related family of teleosts (Gadidae). Unlike other members of the order Gadiformes, which retain R3-generated *aqp3a* and *-3b* paralogs, the Gadidae family encode an additional *aqp3b* gene (Appendix A). Detailed analyses of the two *aqp3b* loci in the GCA_010882105 Celtic Sea assembly of the Atlantic cod genome revealed that the two paralogs are inverted tandem duplicates on LG 4, flanked by three pseudogenes of each paralog (Appendix A). Despite the fragmented nature of the pseudogenes, a Bayesian analysis showed that each clusters within the Gadidae family cluster. These observations indicated that the genes and pseudogenes may have arisen through a combination of unequal crossover events in the ancestor of Gadidae with subsequent deletion remodeling. We named the forward-coding paralog and pseudogenes *aqp3b1* and *aqp3b1p1, -3b1p2, -3b1p3*, respectively, and the reverse-coding paralog and pseudogenes *aqp3b2* and *aqp3b2p1*, *-3b2p2*, *-3b2p3*, respectively.

The third non-WGD lineage-dependent duplication was detected in African and New World cichlids (Cichliformes). In this instance, the two groups of freshwater fishes were separated ~100 Ma in association with the Gondwanan vicariance [71] and both retain duplicated *aqp3a*-type paralogs (*aqp3a1* and *-3a2*) that are tandemly arranged but inverted in the same genomic loci downstream of *nol6* (Appendix A). The phylogenetic signal in this latter tree nevertheless showed the two *aqp3a1* and *-3a2* paralogs of the New World Midas cichlid (*Amphilophus citrinellus*) cluster together as a sister branch to the *aqp3a2* channels of the African cichlids, rather than as separate members of the African cichlid *aqp3a1* and *-3a2* clusters. This indicated that the genes may have duplicated independently in the two lineages, or, alternatively, the New World cichlids *aqp3a* paralogs experienced concerted evolution associated with gene conversion events. To test this hypothesis, we conducted more detailed analyses of 25 cichlid *aqp3a*-type channels. This confirmed the previous topology with New World cichlid *aqp3a1* and *-3a2* channels clustering as a sister branch to the African cichlid *aqp3a2* channels (pp = 1.0) (Appendix A). The more detailed syntenic analyses further confirmed the conservation of the loci but showed that multiple copies of an arrestin domain-containing 3-type protein (*arrdc3*) are differentially amplified adjacent to each *aqp3a* paralog and a piggybac-type transposable element identified in one species (Appendix A). Tests for gene conversion events using “Geneconv” [72] did not reveal any significant inner or outer fragments in the New World cichlid sequences but did reveal a significant a global inner fragment (Bonferroni-corrected Karlin-Altschul *p*-values: 0.009–0.05) corresponding to exon 3 (bp 232–336) that was copied between the *aqp3a1* and *-3a2* paralogs of the African cichlids. Taken together, these data indicated that unequal crossover, transposition, and gene conversion likely acted independently in the two cichlid lineages to give rise to the segmentally duplicated *aqp3a1* and *-3a2* loci after the Gondwanan vicariance.

As stated in the previous section, the fourth non-WGD lineage-dependent duplication occurred as a tandem duplication of the *aqp10* channels to form *aqp10aa* and *-10ab* paralogs in the common ancestor of all actinopterygian fishes. In this case, we uncovered extant *aqp10ab* tandem duplicates in clupeid teleosts, together with orthologous pseudogenes in salmonid genomes (Appendix A). The syntenic data (Appendix A) showed that the teleost *aqp10ab* genes are still tandemly arranged downstream of the *aqp10aa* paralogs. The data also indicated that functional copies of *aqp10ab*, which are expressed in the transcriptome of the Japanese grenadier anchovy (*Coilia nasus*), still exist in some clupeids but are either fractionated as in salmonids or lost in other lineages. The previously named teleost *aqp10b* genes were thus renamed *aqp10bb* to reflect the R3-generated duplicates of the downstream *aqp10ab* channels, while searches for the R3-generated duplicates of the upstream *aqp10aa* genes (previously named *aqp10a*) successfully uncovered the *aqp10ba* genes tandemly arranged upstream of the *aqp10bb* paralogs. These observations revealed that actinopterygian *aqp10* channels experienced a similar duplication history to the actinopterygian *aqp8* channels where tandem duplication preceded the R3 event [35].

A surprising observation, however, was the existence of full-length and apparently functional *aqp10ba* genes encoded in the genomes of seven families of gadiform teleosts (Gadidae, Lotidae, Merlucciidae, Moridae, Phycidae, and Macrouridae) but not in the genomes of closely related stylephoriform or zeiform teleosts, which, together with the Gadiformes, comprise the Zeogadaria. We nevertheless identified complete *aqp10ba* forms in the genome of an aulopiform teleost, the Atlantic greeneye (*Parasudis fraserbrunneri*), and an ostariophysan teleost, the electric eel (*Electrophorus electricus*). In the latter species, the syntenic data showed that the downstream *aqp10bb* paralog, which is the functional channel in most teleosts, is a pseudogene. More detailed analyses of the *aqp10* loci in salmonids revealed that although most species have fractionated the *aqp10ba* channels into pseudogenes, some coregonid species, such as the common whitefish (*Coregonus lavaretus*) and balchen (*Coregonus sp*.), still retain full-length orthologs and an apparently functional form is expressed in the transcriptome of the Tsinling lenok trout (*Brachymystax lenok tsinlingensis*) (Appendix A). Our findings thus showed that an R3-generated duplicate of an ancient *aqp10* tandem duplicate that originated >375 Ma in the common ancestor of extant actinopterygian fishes is lost in most teleost lineages but is still broadly retained in the gadiform order of teleosts that diverged ~300 million years later at the beginning of the Paleogene era [61].

The fifth non-WGD lineage-dependent duplication was identified in the genomes of Atlantic and Pacific eels (Anguillidae). Here, a separate Bayesian analysis of *aqp10bb*-type genes in eight species of elopomorph teleosts revealed that tandem duplication probably (pp = 0.91) occurred in the common ancestor of the Anguillidae family to form *aqp10bb1* and *-10bb2* paralogs, and then it occurred again (pp = 0.81) in the Atlantic eels to form a third *aqp10bb3* paralog (Appendix A).

Finally, the sixth non-WGD lineage-dependent duplication occurred for *aqp10aa*-type genes within submembers of the Atherinomorphae, a superorder of the Acanthomorphacaea (Figure 3; Appendix A). The data indicated that *aqp10aa* genes tandemly duplicated to form *aqp10aa1* and *-10aa2* in a suborder of the Cyprinodontiformes, the Cyprinodontoidei (platyfishes, swordtails, mosquitofishes, sailfins, guppies, mummichogs, and pupfishes) but not in the other suborder, the Aplocheiloidei (killifishes and rivulines). To verify this observation, we re-examined the *aqp10aa*-type Glps assembled from the genomes of 31 species that encompass all three orders of the monophyletic Atherinomorphae, the Atheriniformes, Beloniformes, and Cyprinodontiformes [73]. The new analyses confirmed the original findings (pp = 1.0) showing that tandemly arranged *aqp10aa1* and *-10aa2* genes exist in all sampled members of the Cyprinodontoidei but not in the genomes of the Aplocheiloidei, Beloniformes, or Atheriniformes (Appendix A). It is not yet possible to state the origin of the event, since the data were only sampled from the genomes of New World Cyprinodontoidei. It is nevertheless clear that the upstream *aqp10aa1* genes display accelerated evolutionary rates compared to the downstream *aqp10aa2* paralogs, with the higher mutation rates resulting in pseudogenes in several species.

### 3.5. Glp Evolution in Paleotetraploid Teleosts

The above findings for Teleostei primarily addressed the *glp* repertoires of diploid species. For the teleost lineages that experienced R4, the data for allotetraploid cyprinids were fully consistent with the previous reports for the common carp (*Cyprinus carpio*) in which the *glp* copy number is double (13–14 paralogs) that of zebrafish (7 paralogs) [68,74]. In the present context, however, the *aqp10* genes were renamed *aqp10aa* and *-10bb* in diploid cyprinids and *aqp10aa1*, *-10aa2*, *-10bb1* and *-10bb2* in allotetraploid cyprinids. The consequence of R4 in paleotetraploid salmonids, which occurred ~70–90 million years before the allotetraploid event in cyprinids [45,46,47], is more complex because it is compounded by additional lineage-specific *glp* duplications.

The first lineage-specific duplication was identified amongst *aqp3a*-type genes. In this instance, a tandem duplicate exists in the genome of the northern pike (*Esox lucius*), a closely related protacanthopterygian species (Esociformes) that did not experience R4, while two sister clusters consistent with R4 exist in the genomes of all Salmoniformes (Appendix A). Interestingly, subsequent searches in the downstream loci of eleven salmonid genomes revealed fragmented pseudogenes in several species. The Bayesian inference of the fragmented genes showed that they sort into each of the sister clusters, with the long-branch length of the Arctic charr (*Salvelinus alpinus*) *aqp3a2b* pseudogene caused by a frameshift (Appendix A). The syntenic data further revealed that the locus of each pseudogene is consistent with an ancestral tandem duplication event antecedent to R4 (Appendix A). Though the phylogenetic data indicated that the tandem duplication events may have occurred independently in Esociformes and Salmoniformes, for simplicity, we named the pike *aqp3a*-type genes *aqp3a1a* and *-3a1b* to reflect the upstream and downstream loci, respectively, and the binary clusters of R4-derived *aqp3a*-type genes in salmonids *aqp3a1a-aqp3a1b* and *aqp3a2a-aqp3a2b*.

Amongst *aqp3b*-type genes, we did not find any orthologs in Esociformes, but we identified two clusters in salmonids (*aqp3b1* and *-3b2*) that are consistent with R4 (Appendix A). The data showed that most salmonids retain complete *aqp3b1*-type channels but have fragmented the *aqp3b2*-type channels into pseudogenes. Exceptions are the Arctic charr and the balchen, which appear to have retained complete forms of *aqp3b2*. The differential fragmentation of R4 duplicates in salmonids was also found for the *aqp7* genes, where most species retain one complete copy (*aqp7_1*), while the R4 duplicate (*aqp7_2*) is either lost or fragmented into a pseudogene (Appendix A). We nevertheless observed that some species in the genera *Hucho* and *Brachymystax* appear to have retained complete R4-derived copies of *aqp7_2*. This observation was validated by the expression of both genes in the transcriptome of the Tsinling lenok trout. Conversely, the genomic assemblies indicated that the balchen has tandemly duplicated the *aqp7_1* gene, where the upstream paralog (*aqp7_1a*) now comprises a pseudogene. The *aqp7* ortholog in the genome of the northern pike is also a pseudogene, but this is not representative of the order Esociformes, since a complete form is expressed in the transcriptome of the eastern mudminnow (*Umbra pygmaea*). For *aqp9*-type channels, the data for the salmonids were consistent with R3 and R4, with most species retaining three complete *aqp9a1*, *-9b1* and *-9b2* genes but either losing or retaining a fractionated copy of the *aqp9a2* channel (Appendix A). Again, as an exception, the balchen appears to encode a complete form of the *aqp9a2* gene, while losing the *aqp9a1* paralog.

The salmonid *aqp10*-type channels were the most complex to unravel. This was partly due to the ancient *aqp10aa-aqp10ab* tandem duplication in the common ancestor of all actinopterygian fishes, and the ensuing R3 and R4 WGDs experienced by salmonid genomes. It was also due to gene fragmentation and/or loss together with the translocation of derivative binary gene clusters in salmonids. If the teleost *aqp10* subfamily evolved in a similar manner to the *aqp8*-type channels [35] with all descendent genes surviving, we could expect four genes localized in two chromosomal binary clusters in diploid teleosts (*aqp10aa-aqp10ab* and *aqp10ba-aqp10bb*) and eight genes in four chromosomal binary clusters in paleotetraploid salmonids (*aqp10aa1-aqp10ab1*, *aqp10aa2-aqp10ab2*, *aqp10ba1-aqp10bb1*, and *aqp10ba2-aqp10bb2*). The initial searches in the diploid northern pike genome revealed that the *aqp10aa* and *aqp10bb* genes are localized on separate chromosomes (Appendix A). Subsequent searches in the respective downstream and upstream loci further identified fragmented genes consistent with the above evolutionary hypothesis. Bayesian and syntenic analyses of the downstream *aqp10ab* pseudogene recovered the expected topology (pp = 1.0) in relation to complete orthologs in Clupei and fragmented orthologs in Salmoniformes (Appendix A). We further identified two fragments from exon 3 (15 bp) and exon 5 (51 bp) of the putative *aqp10ba* gene in the expected syntenic locus upstream of the *aqp10bb* channel. (Appendix A). Despite the short length, Bayesian inference recovered the expected topology (pp = 0.82) of the pike *aqp10ba* pseudogene in relation to the *aqp10ba2* pseudogenes of Salmoniformes (Appendix A).

Using the same strategy for the salmonid genomes, we identified three species—the coho salmon (*Oncorhynchus kisutch*), brown trout (*Salmo trutta*), and huchen (*Hucho hucho*)—that harbor seven of the eight predicted genes localized in binary clusters on four separate chromosomes (Appendix A). However, in four other species (the rainbow trout (*Oncorhynchus mykiss*), arctic charr, Atlantic salmon, and grayling (*Thymallus thymallus*), the *aqp10aa2-aqp10ab2* binary clusters are localized on the same chromosomes as the *aqp10aa1-aqp10ab1* clusters. The nucleotide space separating the clusters varies from ~392 kb in the grayling to ~4900 kb in the Atlantic salmon, and the colocalization occurs both downstream in the rainbow trout genome or upstream in the charr, salmon, and grayling genomes. In each case, the colocalization has occurred together with the loss of a chromosomal region that would represent the syntenic paralogon. In the balchen genome, we could only identify two chromosomal regions harboring the *aqp10aa1-aqp10ab1* and *aqp10ba2-aqp10bb2* binary clusters. Conversely in the grayling genome, a second *aqp10bb1b* gene exists ~10,155 kb downstream of a *aqp10bb1a* pseudogene, which is localized in the canonical syntenic position. The above observations indicated that the regions of the salmonid genomes that encode *aqp10*-type channels have experienced differential rearrangement in the aftermath of R4. This is consistent with bursts of DNA transposon activity coinciding with species divergence of salmonids [75]. Indeed, searches for such DNA transposons uncovered >20 DTSsa15-like Tc1/mariner type transposons distributed throughout the length of the Atlantic salmon chromosome 2. Interestingly, two of the DTSsa15-like transposons are localized close to the *aqp10aa1-aqp10ab1* and *aqp10aa2-aqp10ab2* binary clusters (Appendix A).

A final anomaly associated with the aqp10-type genes of salmonids is the lack of lineage sorting of the *aqp10aa1* and *-10aa2* channels and the corresponding *aqp10ab1* and *-10ab2* pseudogenes that would be expected if they were derived from the R4 event. For species that retain complete forms of both copies (including the brown trout, Atlantic salmon, huchen, and grayling), the *aqp10aa1* and *-10aa2* channels cluster on the same branch as the species or genera rather than in separate clusters, with each species or genera represented as per the *aqp10bb1* and *-10bb2* channels (Appendix A). The primary reason for this is the very high conservation of the respective CDS within each species. For example, the nucleotide identities range from 97.7% for the grayling channels to 98.9% for the salmon and trout channels. The sequence conservation of the deduced proteins is even higher, with up to 99.1% identity for the brown trout *aqp10aa1* and *-10aa2* channels. This is despite being encoded in polymorphic gene structures that, respectively, vary in length from 7941 to 8610 bp. These findings suggest that the salmonid aqp10aa-aqp10ab type channels experienced concerted evolution. Using Genconv, we found statistical evidence (Bonferroni-corrected Karlin-Altschul *p*-values: 0.0002–0.05) for short (74–83 bp) global inner and (21–170 bp) global outer fragments for the grayling and huchen genes but not large-scale evidence of major gene conversion events.

### 3.6. Functional Glp Repertoires in Paleotetraploid Salmonids

Given the convoluted history and high copy numbers of *glps* in paleotetraploid salmonids, we conducted two experiments to assess which genes express functional paralogs. Using the Atlantic salmon as the model organism, we isolated eleven *glp* sequences and determined their orthologies, loci, gene structures, and coding directions based upon the extensive phylogenetic and syntenic analyses described in the previous sections (Figure 4A, Appendix A). We then examined their tissue-wide expression patterns in four females and four males via RT-PCR, and we expressed each cRNA in *X. laevis* oocytes to evaluate their permeation properties for water and glycerol. Of the eleven sequences isolated, nine are complete paralogs (*aqp3a1a*, *-3a2a*, *-3b1*, *-7_1*, *-9a1*, *-9b1*, *-9b2*, *-10aa1* and *-10bb1*) and the remaining two are isoforms: a deletion variant of *aqp9a1* (*aqp9a1_v2*) lacking exon 3, and an M3 variant of *aqp10aa1* (*aqp10aa1_v2*) with a 39 amino acid shorter N-terminus. Attempts to isolate *aqp10aa2* were not successful.

The tissue-wide expression profiles did not reveal major differences between the sexes, except for *aqp9b1* and *-9b2*, which are expressed in the lens of males but not females (Figure 4B). In addition, some differences were noted for *aqp3a1a* in the female ureter and *aqp3b1* in the male rectum, ureter, and testis. Near ubiquitous expression profiles are seen for *aqp7_1* and the two isoforms of *aqp9a1*, while greater selectivity is seen for the tissue expression patterns of the other channels. For example, *aqp10aa1* shows the most restricted profile in the pyloric caeca and liver, and *aqp10bb1* expression is restricted to post gastric regions of the intestine. Conversely, the *aqp9b1* and *-9b2* channels are more selectively expressed in neural and visual tissues, while the *aqp3*-type channels are expressed in osmoregulatory tissues such as gills, pseudobranchs, skin, esophagus, stomach, mid-intestine, rectum, and ureter.

The functional assays in *X. laevis* oocytes demonstrated that all channels are highly significantly (*p* < 0.001) permeable to water and glycerol, except the Aqp9a1_v2 and Aqp10aa1_v2 deletion variants, which permeate neither water nor glycerol (Figure 5). These experiments thus revealed that of the 20 *glps* that evolved in the order Salmoniformes, at least nine are functional in Atlantic salmon, which has further extended its *glp* repertoire through splice variation. In this latter instance, it seems possible that an *aqp9a1_v2* exon 3 deletion variant could play a dominant-negative regulatory role, since it is widely expressed in the same tissues as the complete *aqp9a1_v1* isoform but it is not functional when expressed in *X. laevis* oocytes. Based upon the identification of complete orthologs in other members of the order Salmoniformes, the copy number of functional *glp* paralogs was found to range from nine in charrs, coregonid whitefishes, and Pacific salmons to eleven in the huchen. Such diversity is less than the thirteen-to-fourteen paralogs of tetraploid cyprinids, but it is close to twice the prevalence of orthologs in diploid teleosts. The above analyses further revealed that the multiplicity of *glps* in Salmoniformes mostly derives from WGD rather than tandem duplication, with neofunctionalization occurring at the expression level rather than channel selectivity.

### 3.7. A Timeline of Vertebrate Glp Evolution

To summarize the evolution and newly derived nomenclature of *glps* in vertebrates, we annotated the identified gene duplication, fractionation, and/or loss events in relation to the divergence times of each investigated lineage (Figure 6A,B). The scheme shows the influence that four rounds (R1–4) of WGD, three separate interchromosomal duplications or translocations, and >20 tandem duplication events had on the evolution of the vertebrate *glps*. It further shows that gene copy numbers, although similar in several lineages, have different origins. For example, five *glp* genes exist in Elasmobranchii, Polypteriformes, Acipenseriformes, Anura, and Caudata, but in each lineage, independent tandem duplication events explain the disparate origins of the *aqp3C2* and *-10C2* channels in the elasmobranchs, the *aqp10ab* channels in the actinopterygian fishes, and the *AQP7* and *-13* channels in the amphibians.

To estimate the timing of the different duplication or loss events, we integrated our findings with divergence time estimations in the literature [61,76,77]. The specific timing of a number of events nevertheless remains obscure due to the absence of the gene or genomic data. For example, the loss of *aqp13* genes in the chondrichthyan, actinopterygian, and sarcopterygian lineages is inferred at the root of Chondrichthyes and Actinopterygii but is more arbitrarily annotated in the sarcopterygian lineage. The same is true for the loss of *AQP10* genes in Testudines and Prototheria. Conversely, the estimates of gene duplications in Petromyzontiformes were placed in accordance with the divergence times of the sampled *Lentheron* and *Petromyzon* genera [76]. Similarly, the *aqp10bb1* and *-10bb2* duplication in Anguillidae, as well as the *aqp3b1* and *-3b2* duplication in Gadidae, were placed in accordance with the divergence times of the respective families [61]. To assess the timing of the *aqp10aa1* and *-10aa2* duplication in Cyprinodontoidei and the *aqp3a1* and *-3a2* duplication in African Cichliformes, we performed independent Bayesian timetree analyses of the two events implemented with 100 million MCMC generations in BEAST under strict and relaxed clock conditions. The results indicated that the *aqp10aa1* and *-10aa2* duplication in Cyprinodontoidei occurred between 42 and 53 Ma, with a maximum difference of 8.3 million years separating the nodes under strict clock conditions (Figure 6C). This timing is close to the ~50 Ma divergence time estimation of the group [61]. Conversely, the *aqp3a1* and *-3a2* duplication in African Cichliformes occurred 11–16 Ma, with a maximum difference of 2.4 million years separating the nodes under strict clock conditions (Figure 6D). This latter estimate was lower than the ~24 Ma age of the last common ancestor of the Oreochromini, Lamprologini, Haplochromini, and Tropheini within the African clade of cichlids [71].

## 4. Discussion

The present work provides a reasonably comprehensive overview of the evolution of glycerol transporters in deuterostome organisms. The findings showed that gene expansion in Ambulacraria was primarily driven by tandem duplication that generated three major *glp* subfamilies (*glp1*, *-2*, and *-3*), while non-vertebrate cephalochordate and tunicate lineages mostly evolved single copy *glp* genes. Tandem duplication was further responsible for lineage-specific expansions within each of the Ambulacrarian *glp* subfamilies, leading to increased repertoires of *glp3*-type channels in hemichordate acorn worms, *glp2*-type channels in ophiuroid brittle starts, and *glp1*-type channels in echinoderm sea urchins, sea cucumbers, and starfishes. A surprising feature of the evolution of vertebrate *glps* is that although serial rounds of WGD can explain much of the observed channel diversity, tandem duplication also played an important role in shaping the genomic repertoires.

Previous analyses of the origins of the vertebrate *glp* subfamilies have indicated that they likely originated in a narrow window time in association with WGD [12,35,78]. However, a confounding aspect of the evolution of these channels in vertebrates has been the observation that certain lineages, including prototherian mammals and anuran frogs, harbor a fifth *glp* gene—the *AQP13* channel [34,35]. The current analyses confirmed the existence of *AQP13* in Prototheria and Amphibia but further provided phylogenetic support for its existence in hagfishes (Hyperotreti). Though the latter support could be caused by long-branch attraction, the extensive codon analyses consistently placed one of the two hagfish sequences in an ancestral position of the *AQP13* cluster, irrespective of the taxonomic sampling, while the other sequence tended to cluster between the *AQP13* and *AQP3* channels. This indicated that the *AQP13* subfamily is ancient and potentially one of the founding members of the vertebrate *glps*. The data for lamprey (Hyperoartia) showed that the two *aqp3*-like sequences consistently cluster with gnathostome *aqp3*, while the other *aqp9_13*-like sequences, which evolved through tandem duplication, either cluster with gnathostome *aqp9* or between the *aqp3* and *aqp13* channels. Conversely, the gnathostome channels robustly resolve into the five *glp* subfamilies (*aqp3*, *-7*, *-9*, *-10*, and *-13*). If we apply the one-to-four rule for two rounds of WGD [79], only four of these subfamilies can be explained by WGD. Since there are currently >1000 vertebrate genomes sequenced and none show evidence of a sixth *glp* subfamily, it seems likely that one of the gnathostome subfamilies evolved through a different mechanism.

The best candidate would seem to be *AQP7*, which is tandemly arrayed downstream of *AQP3* in all Sarcopterygian genomes. However, as revealed by the *glp1b* genes in asteroids, which are juxtaposed to the *glp3* channels, a tandem locus may not be evidence of the genes duplication origin. Furthermore, until now, it has not been possible to provide extensive evidence for the linkage between *aqp3* and *aqp7* genes in actinopterygian fishes, nor a similar linked gene in cartilaginous fishes (Chondrichthyes). The data for basal lineages of actinopterygian fishes now show the linkage of *aqp7* to the *aqp3* paralog in Cladistia and Chondrostei and to the *aqp3b* paralog in Otomorphan teleosts. We further uncovered a tandem duplicate downstream of the *aqp3* paralog in all lineages of cartilaginous fishes. However, in this latter instance the phylogenetic data revealed that the downstream paralog is not an *aqp7*-type channel but a second form of *aqp3* (*aqp3C2*). We therefore found no evidence for the existence of *aqp7*-type channels in any vertebrate lineage prior to the evolution of true bony fishes (Osteichthyes). It thus seems reasonable to propose a new scheme for the evolution of *glps* in Deuterostomia (Figure 7). The scheme combines the observations in this study with the divergence times of the different lineages [77], and it shows that *glps* indeed likely arose during a narrow window of time through tandem duplication in Ambulacraria, and WGD together with tandem duplication in Chordata.

The above interpretation requires the loss of the *aqp13* paralog in most of the vertebrate lineages, yet its survival in Hyperotreti, Hyperoartia, Amphibia, and Prototheria. While such haphazard gene survival may seem unlikely, due to the ~250 million years separating the last common ancestor of the four lineages [80], it is nevertheless the pattern of *aqp10ba* evolution observed in the Gadiformes order of teleost fishes. In this latter instance, we also uncovered complete orthologs of the *aqp10ba* channel in the ostariophysan (electric eel), protacanthopterygian (selected salmonids), and aulopiform (Atlantic greeneye) lineages. Our models showed that the *aqp10ba* channel arose as an R3 duplicate of *aqp10aa* ~300 Ma, and although it was lost in the majority of teleost lineages, it still survives in Gadiformes, which diverged ~225 million years after the R3 event. The proposed scheme of *glp* evolution in Chordata also implies that the lamprey *aqp9_13L* tandem duplicates are in fact *aqp13* channels. While our phylogenetic data did not demonstrate this facet, it seems possible that over the ~475 million years of evolution since the separation of the Petromyzontiform order of lampreys and the Myxiniformes order of hagfishes [76,81], the *aqp9_13L* channels could have convergently evolved molecular features of the gnathostome *aqp9* channels. As in our previous analysis of the aquaporin superfamily in vertebrates [35], the present phylogenetic data for the four lamprey *glps* did not support their R2 origin, which has been suggested to have occurred before the divergence of cyclostomes and gnathostomes [82,83]. On the contrary, it rather supported their lineage-specific interchomosomal and tandem duplications within petromyzontiform lampreys.

While the timing of the R2 WGD in the chordate lineage continues to be debated [84,85], its effect on gnathostome gene and genome evolution is less controversial [41,86,87,88]. Similarly, the timing and effects of WGD in the teleost lineage are well documented [42,89] and consistent with the evolution of the *glp* channels examined here. It is nevertheless recognized that genome reduction (rediploidization) is a dominant mode of evolution [49]. Consequently, it is surprising to note the asymmetric gene evolution of *glp* genes in actinopterygian fishes compared to tetrapods. Both lineages evolved over similar time scales (~400 million years), but tetrapods have mostly retained the same four paralogs (*AQP3*, *-7*, *-9*, and *-10*) with comparatively little gene loss. The main exception is the *AQP10* gene, which appears to be lost in Testudines and Prototheria and is a pseudogene in Ruminantia [70], several families within Muroidea [69], and some species of Metatheria. By contrast, we found substantial evidence of multiple *glp* gene losses in actinopterygian fishes, with a loss of *aqp7* in Holostei and *aqp7b* in Teleostei, *aqp10ba* and *-10ab* in most lineages of Teleostei, *aqp3a* and *-9b* in Osteoglossiformes, *aqp3b* and *-10aa* in Elopomorpha, and *aqp3b* in many orders of Percamorphaceae. Similarly, non-WGD *glp* expansions in Tetrapoda are rare, with only a few instances of tandem duplication at the individual species level. One exception is at the terminal branches of primate evolution, where tandem duplication of canonical *AQP7* occurred at least as far back (~10–13 Ma) in the last common ancestor (LCA) of Homininae [90] but continued to duplicate in chimpanzees and humans. Subsequently, genomic rearrangements reduced the repertoire of functional *AQP7* channels in humans, leaving six pseudogenes spread across two chromosomes. Such differences may be related to the fruit and vegetable diet of gorillas and chimpanzees compared to the more omnivorous diet of humans [91]. Amongst Actinopterygii, and indeed the more basal lineages of Chondrichthyes and Hyperoartia, we found ample evidence of lineage-specific gene retention of additional *aqp3* and *-10* channels in the aftermath of tandem duplication. In the absence of detailed phylogenetic and syntenic analyses, such homologs could confound BLAST searches aimed at identifying WGD duplicates or single-copy orthologous sequences for divergence time estimations. For example, our analyses revealed that even for relatively recent duplications such as the *aqp3a1* and *aqp3a2* channels in Cichliformes and the *aqp10aa1* and *aqp10aa2* channels in Cyprinodontoidei, divergence time estimations could differ by millions of years. These observations highlight the complex challenges facing evolutionary investigations of actinopterygian fishes.

The paleotetraploid Salmoniformes are an order of teleosts that clearly represent such a challenge. Our new estimate of the ancestral *glp* gene copy number was 20 paralogs, of which approximately half are functional. While this is consistent with rediploidization and a differential retention of post-R4 paralogs compared to the R3 paralogs [92], our data revealed unexpectedly complex genomic rearrangements, even in closely related paralogs. The strange case of the *aqp10aa1* and *aqp10aa2* channels indicated that they not only experienced concerted evolution post R4 but have been differentially translocated to become linked in several species, including the rainbow trout (*Oncorhynchus*), arctic charr (*Salvelinus*), Atlantic salmon (*Salmo*), and grayling (*Thymallus*) but not in other congeners such as the coho salmon (*Oncorhynchus*) or the brown trout (*Salmo*). Assuming that the genome assemblies are correct, then these observations are consistent with the bursts of DNA transposon activity including the DTSsa15-like Tc1/mariner type identified in the present study coinciding with speciation events [75]. However, since we predict that the *aqp10aa2* genes are products of R4 as indicated by the syntenic analyses, the absence of large-scale gene conversion events seems paradoxical. The sequence homogenization is not restricted to the *aqp10aa1* and *-10aa2* genes, as it is also evident in the downstream *aqp10ab1* and *-10ab2* pseudogenes, which evolved ~300 million years earlier by tandem duplication in the LCA of Actinopterygii. The observed concerted evolution may therefore be associated with the transposon-mediated interlocus transposition of the binary clusters rather than canonical gene conversion. It is nevertheless clear that the *aqp10aa1* gene displays the most restricted tissue expression profile of all of the salmonid *glps*, even compared to the *aqp10bb1* channel [93], indicating that despite the absence of differentiation in the coding regions, the cis-regulatory regions have nonfunctionalized.

## 5. Conclusions

In conclusion we conducted broad level phylogenetic and syntenic analyses to reconstruct the evolutionary history of glycerol-transport genes (*glps*) in Deuterostomia. The data encompassed the genomic repertoires of all phyla including 41 Echinodermata, two Hemichordata, and 515 Chordata, and they included searches of >1000 deuterostome genomes to verify that all subfamilies were identified. The data revealed that tandem duplication played an unexpectedly important role for the generation of novel channels in nearly all lineages examined. Tandem duplication was thus the main driver of channel diversity early in the evolution of echinoderms and hemichordates, and it remained the major mechanism of subsequent gene expansions in these lineages. Our models for channel duplication in chordates were consistent with serial rounds of WGD combined with tandem duplication, and they indicated that *aqp3* and *-13*-like channels were the likely stem *glp* subfamilies in craniates. However, by integrating analyses of >100 pseudogenes, we revealed how gene fractionation, gene loss, and new tandem duplications resulted in asymmetric *glp* gene evolution in gnathostome lineages, and we showed how such asymmetries can impact estimations of divergence times by millions of years. Rediploidization mechanisms had a greater effect on reducing the *glp* gene repertoires of actinopterygian fishes, although as in cartilaginous fishes, such reductions were compensated for by lineage-specific tandem duplications. The data thus showed that gene copy numbers are poor predictors of the orthologous relationships. We found evidence of independent *glp* gene translocations, gene conversion, and concerted evolution in humans, teleosts, and star fishes, with Tc1/mariner type DNA transposons likely causing the complex rearrangements observed in paleotetraploid salmonids. The tissue-wide expression patterns and experimental tests of the permeation properties of the salmon channels demonstrated that of the 20 paralogs that represent the ancestral condition, approximately half are functional. These data further showed that neofunctionalization has occurred at the transcriptional level rather than within the biophysical properties of the proteins. Based upon the extensive analyses outlined here, we propose a new nomenclature for *glp* genes in Deuterostomia.

## Figures and Tables

**Figure 1 cells-09-01663-f001:**
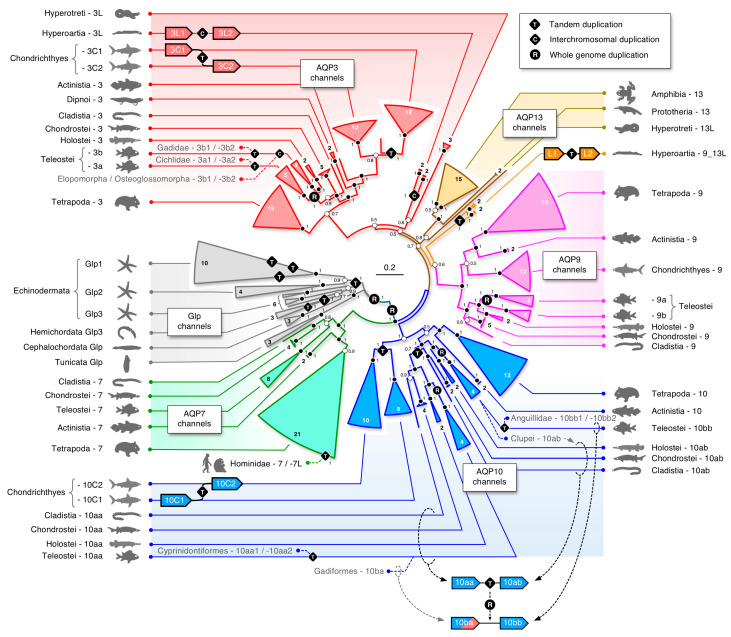
Molecular phylogeny of deuterostome glycerol transporters. The tree is mid-point rooted and inferred via maximum likelihood (RAxML: generalized time-reversible (GTR) gamma model with 3000 bootstraps and search for the best-scoring maximum likelihood (ML) tree) and Bayesian analysis (15 million Markov chain Monte Carlo (MCMC) generations) of 315,034 nucleotide sites of 249 taxa partitioned by codon. Numbers within or external to each collapsed triangular branch refer to the number of taxa with Bayesian posterior probabilities annotated at each node. The scale bar indicates the expected rate of substitutions per site. Modes of gene duplication are shown on respective branches with additional duplications annotated for Teleostei and Homininae. Circled branches within the *AQP10* channel cluster illustrate the origin of the polygenic *aqp10* subfamily in actinopterygian fishes. (See Appendix A for the fully annotated tree including RAxML bootstrap support values).

**Figure 2 cells-09-01663-f002:**
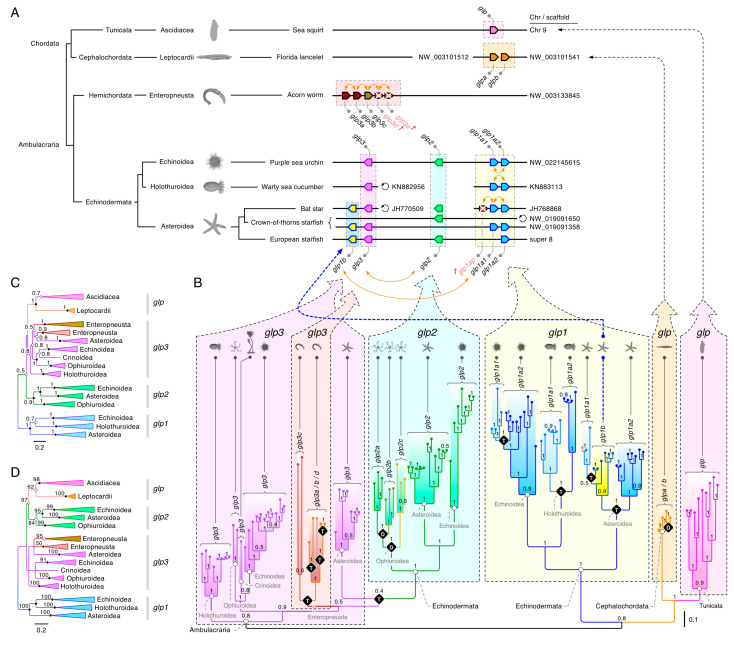
Synteny and molecular phylogeny of non-vertebrate chordate and Ambulacrarian glycerol-transporting channels (*glps*). (**A**) Syntenies are assembled from genome sequences and arranged according to the taxonomic position of the selected organisms. Gene coding direction is indicated by the pointed end of each gene symbol with linkages represented by solid lines between paralogs. A circular arrow indicates that the chromosome (Chr) or scaffold is flipped in relation to the European starfish super 8 scaffold. Orange arrows above gene symbols indicate tandem duplication. (**B**) Mid-point rooted Bayesian majority rule consensus tree (5 million MCMC generations) of 97,721 nucleotide sites of 116 taxa partitioned by codon. Bayesian posterior probabilities are annotated at each node with tandem “T” or unspecified “D” modes of duplication shown on selected nodes. The scale bar indicates the expected rate of substitutions per site. (See Appendix A for the fully annotated tree). (**C**) Mid-point rooted FastTree using GTR model optimized with gamma20 likelihood showing support values at each node. (**D**) Mid-point rooted maximum likelihood tree (PAUP: heuristic search optimized for parsimony GTR model with NST = 6; partitioned by codon) with RAxML support values shown at each node.

**Figure 3 cells-09-01663-f003:**
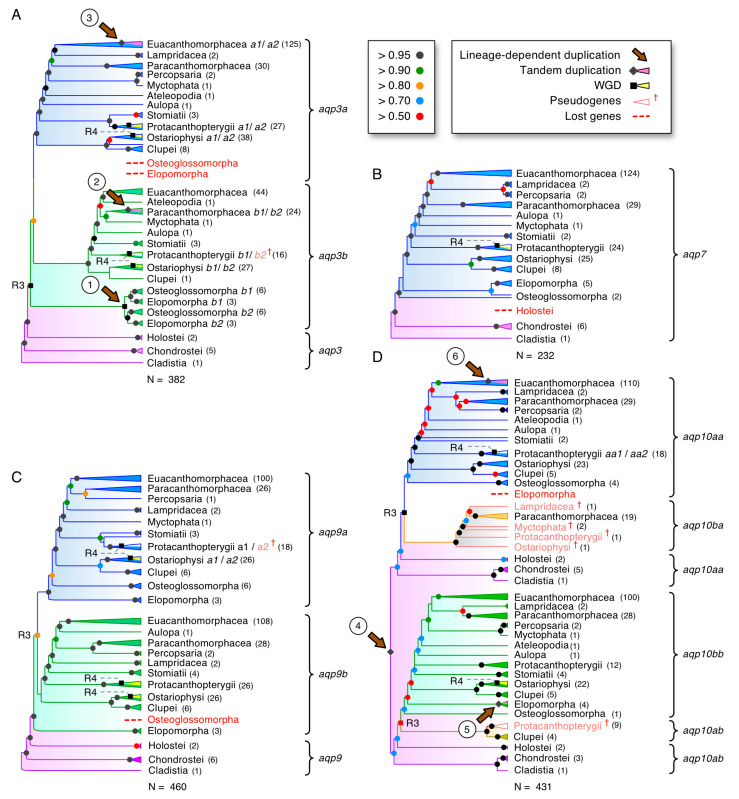
Molecular phylogenies of actinopterygian *glps*. (**A**) Mid-point rooted Bayesian majority rule consensus tree of aqp3 channels (30 million MCMC generations) of 365,143 nucleotide sites of 382 taxa partitioned by codon. (See Appendix A for the fully annotated tree). (**B**) Bayesian majority rule consensus tree of *aqp7* channels (15 million MCMC generations) of 225,618 nucleotide sites of 232 taxa partitioned by codon. The tree is rooted with reedfish (*Erpetoichthys calabaricus*) *aqp7*. (See Appendix A for the fully annotated tree). (**C**) Mid-point rooted Bayesian majority rule consensus tree of *aqp9* channels (40 million MCMC generations) of 440,522 nucleotide sites of 460 taxa partitioned by codon. (See Appendix A for the fully annotated tree). (**D**) Mid-point rooted Bayesian majority rule consensus tree of *aqp10* channels (50 million MCMC generations) of 457,471 nucleotide sites of 431 taxa partitioned by codon. (See Appendix A for the fully annotated tree). Bayesian posterior probabilities are indicated with color nodes as specified in the key. Six non-WGD lineage-dependent duplications are highlighted together with modes of gene duplication or loss in each tree.

**Figure 4 cells-09-01663-f004:**
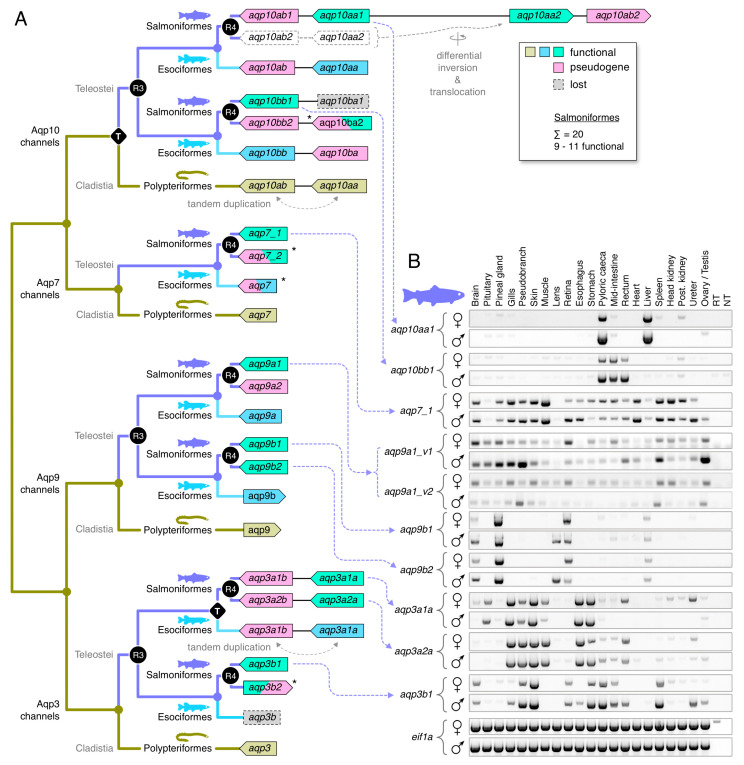
Evolution and expression of *glps* in paleotetraploid salmonids. (**A**) Schematic representation of the evolution, coding direction, and diversification of *glps* in a non-teleostean bichir and protocanthopterygian teleosts (Esociformes and Salmoniformes). Whole genome (R3 and R4) and tandem (T) duplication events are annotated on relevant nodes. (**B**) *glp* tissue expression profiles (RT-PCR) from four male and four female Atlantic salmon using elongation initiation factor 1a (*eif1a*) as a reference. Negative controls: cDNA without reverse transcriptase (RT) or no cDNA (NT).

**Figure 5 cells-09-01663-f005:**
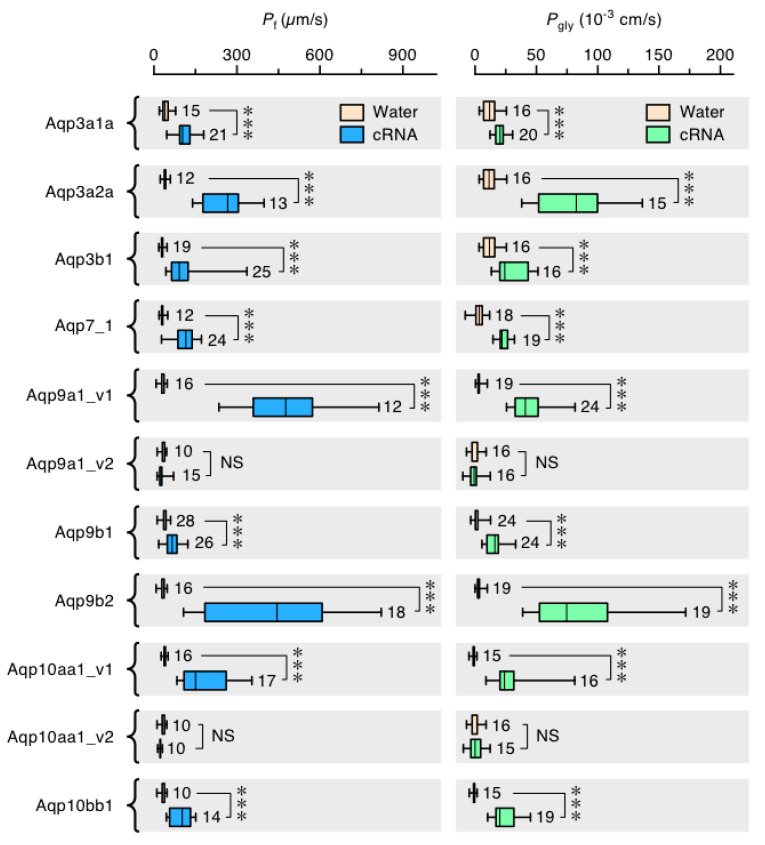
Functional characterization of Atlantic salmon Glps. Osmotic water permeability (*P*_f_) and glycerol uptake (*P*_gly_) of *X. laevis* oocytes injected with water (control) or cRNAs encoding the Atlantic salmon Glps. Data are box and whisker plots with the number of biologically independent oocytes indicated beside each plot. *** *p* < 0.001, statistically different (unpaired Student’s *t*-test), with respect to control oocytes. NS: not statistically significant with respect to control oocytes.

**Figure 6 cells-09-01663-f006:**
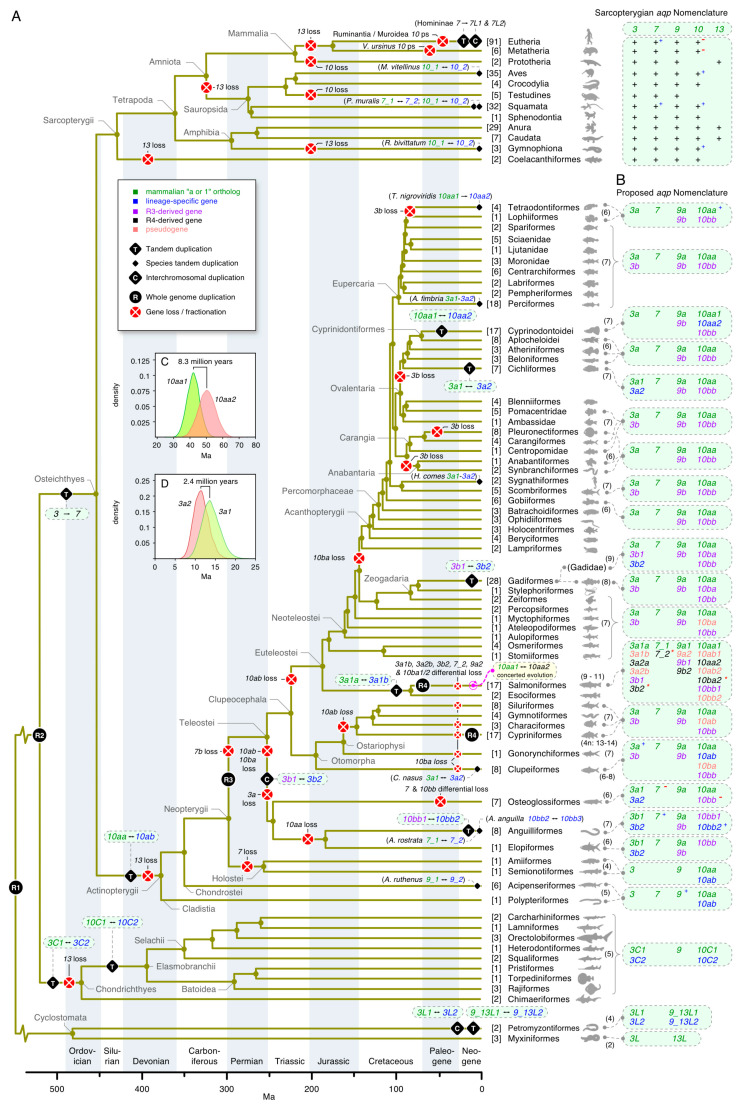
A timeline for the evolution of vertebrate *glps*. (**A**) A schematic representation of historical events that occurred during the evolution of vertebrate *glps*. Data are arranged in accordance with the divergence of each lineage through geological time [61,77]. Ma: millions of years ago. The presence of single copy *aqp3*, *-7*, *-9*, *-10*, or *-13* orthologs in Sarcopterygii is indicated with a “+” with extra copies or loss within each subfamily superscripted “+” (blue) or “-” (red). Four rounds (R1–R4) of whole genome duplication, together with three interchromosomal duplications/translocations and >20 tandem duplication and gene loss events, are shown. Numbers in square brackets refer to the number of taxa sourced for coding sequence (CDS) assembly. (**B**) Proposed aquaporin (*aqp*) nomenclature in non-sarcopterygian vertebrates, with the genes colored according to their origins. Numbers in parentheses refer to gene copy numbers. (**C**) Bayesian divergence time estimation of *aqp10aa1* and *-10aa2* duplication in Cyprinodontoidei. (**D**) Bayesian divergence time estimation of *aqp3a1* and *-3a2* duplication in African cichlids.

**Figure 7 cells-09-01663-f007:**
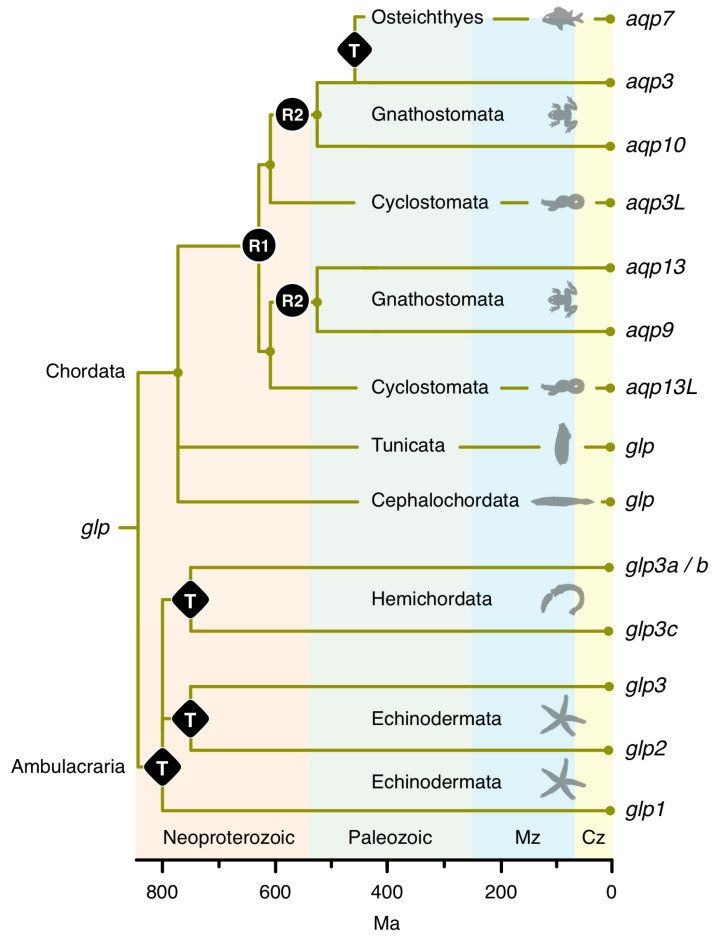
The origin of deuterostome Glp subfamilies. A proposed scheme for the origin of *glp* gene subfamilies in Deuterostomia as a function of geological time. Modes of gene expansion are indicated as T: tandem duplication; or R1 and R2: whole genome duplication. Mz: Mesozoic; Cz: Cenozoic; and Ma: millions of years ago.

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
