# Peer review of "Unravelling the Complex Duplication History of Deuterostome Glycerol Transporters"

_cells, 2020, doi:10.3390/cells9071663_

Round 1

Reviewer 1 Report

The article is very well written. I have very few concerns as listed below. 

Line 1 – Abstract – I guess it should to article

In the title “Unravelling an unexpectedly” – this looks inappropriate

Abstract

“Transmembrane glycerol transport is an ancient biophysical property that evolved in selected subfamilies of water channel (aquaporin) proteins.”  - ancient biophysical property- need to change

“interrogation” – here evolution, analysis or study will be more appropriate. Use of interrogation has different meaning

“We uncover independent examples of gene translocations” – Need improvment

“The combined findings form 37 the novel basis for proposing a pandeuterostome glp gene nomenclature” – Authors can find better conclusion remark at the end of abstract

Introduction

 “superfamily of transmembrane water-conducting channels, the aquaporins” - -Please make sure, AQP is not a superfamily, AQP belong to MIP superfamily.

“through evolutionary time” – can be removed

Results

AQP are known to transport many i=uncharged small solutes. In results or discussion better to address evolutionary aspect for solute specificity more particularly for solutes other than the glycerol.

Authors have not considered conserved motifs which defines solute specificity for instance NPA motifs and Ar/R selectivity filters.

Author Response

The article is very well written. I have very few concerns as listed below.

Line 1 – Abstract – I guess it should to article

Response: We have change the title as follows: “Unravelling the complex duplication history of deuterostome glycerol transporters”

In the title “Unravelling an unexpectedly” – this looks inappropriate

Abstract

“Transmembrane glycerol transport is an ancient biophysical property that evolved in selected subfamilies of water channel (aquaporin) proteins.” - ancient biophysical property- need to change

Response: The use of the word “ancient” is correct in this context

“interrogation” – here evolution, analysis or study will be more appropriate. Use of interrogation has different meaning

Response: The word “interrogation” has been changed to “analyses”

“We uncover independent examples of gene translocations” – Need improvement

Response: This sentence has been changed to: “We uncover separate examples of gene translocations …”

“The combined findings form the novel basis for proposing a pandeuterostome glp gene nomenclature” – Authors can find better conclusion remark at the end of abstract

We have altered the sentence as follows: “The combined findings resolve the origins and diversification of glps over >800 million years and thus form the novel basis for proposing a pandeuterostome glp gene nomenclature.”

Introduction

“superfamily of transmembrane water-conducting channels, the aquaporins” - -Please make sure, AQP is not a superfamily, AQP belong to MIP superfamily.

Response: In 1993 the use of the term “aquaporin” was coined to encompass the superfamily of water channel proteins, see Agre, P.; Sasaki, S.; Chrispeels, M.J. Aquaporins: a family of water channel proteins. Am. J. Physiol. 1993, 265, F461." Hence MIP is an outdated terminology. See also Finn, R.N.; Cerdà, J. Aquaporin. In Encyclopedia of Signaling Molecules. 2nd Edition, Choi, S. Ed.; Springer: New York, 2018; pp. 1-18.

“through evolutionary time” – can be removed

Response: removed

Results

AQP are known to transport many i=uncharged small solutes. In results or discussion better to address evolutionary aspect for solute specificity more particularly for solutes other than the glycerol.

Response: The aim of the study, as explained in the introduction was gene discovery, not solute specificity which would need to be addressed experimentally.

Authors have not considered conserved motifs which defines solute specificity for instance NPA motifs and Ar/R selectivity filters.

Response: This was beyond the scope of the current study, which focused on gene discovery as explained in the point above.

Reviewer 2 Report

The authors carry out an interesting evolutionary study in Deuterostomia and provide a broad overview over the evolution of aquaglyceroporin (Glps). They identify many new Glps in various species and characterise salmon GlPs in more detail including a determination of their transport function.

Major

The quality of the figures is low. Please increase the resolution.

Minor

It is confusing that the authors propose that they studied the function of Glp in paleotetraploid salmonids and refer to Fig. 4A and Supplementary Fig. 14, whereas the functional data are presented in Fig. 5. Please align.

Author Response

The authors carry out an interesting evolutionary study in Deuterostomia and provide a broad overview over the evolution of aquaglyceroporin (Glps). They identify many new Glps in various species and characterise salmon GlPs in more detail including a determination of their transport function.

Major

The quality of the figures is low. Please increase the resolution.

Response: We have requested to upload high-resolution figures

Minor

It is confusing that the authors propose that they studied the function of Glp in paleotetraploid salmonids and refer to Fig. 4A and Supplementary Fig. 14, whereas the functional data are presented in Fig. 5. Please align.

Response: This is now corrected as follows: “Functional Glp repertoires in paleotetraploid salmonids